# Disentangling kinetics from thermo-dynamics in heterogeneous colloidal systems

Hamed Almohammadi [1], Sandra Martinek[1], Ye Yuan[1], Peter Fischer [1] & Raffaele Mezzenga [1,2]

In Nucleation and Growth, the process by which most heterogeneous systems form, thermodynamics sets the asymptotic boundaries toward which the system must evolve, while kinetics tries to cope with it by imposing the transport rates. In all heterogeneous colloidal systems observed in nature, composition, shape, structure and physical properties result from the trade-off between thermodynamics and kinetics. Here we show, by carefully selecting colloidal systems and controlling phase separation in microfluidic devices, that it becomes possible to disentangle kinetics effects from thermodynamics. Using amyloids and nanocellulose filamentous colloids, we demonstrate that decoupling kinetics from thermodynamics in the phase separation process unveils new physical phenomena, such as orders of magnitude shorter timescales, a wider phase diagram, and structures that are not observable via conventional liquid-liquid phase separation. Our approach enables on-demand fabrication of multicomponent heterogeneous liquid crystals, enhancing their potential, and introducing original fundamental and technological directions in multicomponent structured fluids.

Nucleation and growth (N&G)−the emergence of a new phase within an initially homogeneous one−is one of the most important physical phenomena by which gas-liquid (GLPS), liquid-liquid (LLPS) and solid-liquid (SLPS) phase separation takes place. The occurrence of N&G is ubiquitous in the universe and central to many scientific disciplines ranging from physics to material science, biology, and medicine. In solid-liquid phase separation, examples include phenomena as general as the freezing of water into ice[1,2], solidification of a molten metal[3], or formation of crystals in biomineralization[4]. In gas-liquid phase separation, N&G controls the formation of gas bubbles from supersaturated liquids[5], such as $CO_2$ bubbles sparking from a freshly opened bottle of champagne. N&G also plays a vital role in LLPS observed in polymeric fluids and colloidal dispersions; in vivo, it is furthermore associated with important biological processes such as the formation of intracellular membraneless organelles[6–8]. In technology, controlling N&G may allow reducing the timescale of formation of new phases and designing new materials with desired composition, structure, shape, size and ulti-mately physical properties[9–14]; not surprisingly N&G is indeed a process relevant to many fields including food[15], electronic[14,16], photonic[17] and pharmaceutical[18] industries. Yet, controlling N&G is largely limited by kinetic factors, including the stochastic nature of nucleation and the finite transport rates ruling the growth phase[19], so that the time para-meter becomes pervasive in most heterogeneous systems, intimately connecting thermodynamics and kinetics aspects.

Heterogeneous systems based on filamentous biological colloids[20–25] are a class of matter that undergoes N&G via a distinct phase separation mechanism[26,27] and bear both fundamental and tech-nological significance[28]. In this system, as first shown by Onsager[29], the phase separation has a purely entropic origin and stems from the interplay between orientational entropy and excluded volume packing entropy−a phenomenon known as liquid-liquid crystalline phase separation (LLCPS)[26,27] to distinguish it from the more common LLPS

---

[1]Department of Health Sciences and Technology, ETH Zurich, Zurich, Switzerland. [2]Department of Materials, ETH Zurich, Zurich, Switzerland. e-mail: raffaele.mezzenga@hest.ethz.ch

trade-off between enthalpy and entropy. This fundamental thermodynamic difference leads to a dramatic change in their respective phase diagrams: while the binodal line separating the 2-phase from the 1-phase region in LLPS maintains a finite slope[30], in LLCPS the binodal line is represented by two perfectly vertical lines separating the 2-phase region, where isotropic and nematic phases coexist, from the isotropic and nematic single phase regions at low and high colloid composition, respectively[29]. These two vertical binodal branches occur at the Onsager volume fractions, which for monodisperse rods of length $L$ and diameter $D$, would locate at $\phi_I = 3.34(D/L)$ and $\phi_N = 4.49(D/L)$, respectively[29]. In practice, for polydisperse rod systems these two exact limits depend on rod length distribution[31]; for the two systems considered, however, an estimation of these two branches using the average contour length of the rods has been shown to closely predict the experimental values[32,33].

Within the 2-phase composition region (Fig. 1a), phase separation via N&G promotes the formation of microdroplets of nematic phase with high concentration and orientational order within an isotropic phase with no orientational order and low concentration[20–25]. These microdroplets, called tactoids, feature a fascinating rich phase behavior[25,32] and, unlike ordinary spherical colloids that crystallize to a roughly spherical shape, assume spindle-like, prolate or oblate shapes[20–25,32] due to the subtle interplay between the vanishingly small interfacial tension and liquid crystalline anisotropic elastic tensor[25,34], Fig. 1b. Upon growth of the tactoids with time, with N&G time of the order of minutes to days when mechanical disturbance is avoided, their self-selected shape and structure change[20–25,32–34], with volume-composition curves evolving via trajectories of varying slopes depending on the system of interest, but always confined within the two Onsager vertical asymptotes (Fig. 1a). These volume-composition curves, which only recently have been theoretically estimated for rod-like objects undergoing LLCPS[26], directly reflect kinetics effects as they are a consequence of finite transport rates: only for infinitely fast transport, would they perfectly overlap with the two vertical Onsager composition lines. Thus, differently from LLPS, the infinite slope of the two Onsager vertical asymptotes sets a precise limit to the final compositions of the two phases and opens the possibility of manipulating and controlling morphology and structure by relying exclusively on the two equilibrium asymptotic compositions. For example, this could allow disentangling the internal periodicity of cholesteric tactoids, also known as cholesteric pitch, from the tactoid size, a breakthrough which has remained elusive to date[25,26].

In what follows, using β-lactoglobulin amyloid fibrils and sulfated cellulose nanocrystals as two model anisotropic colloidal systems, we

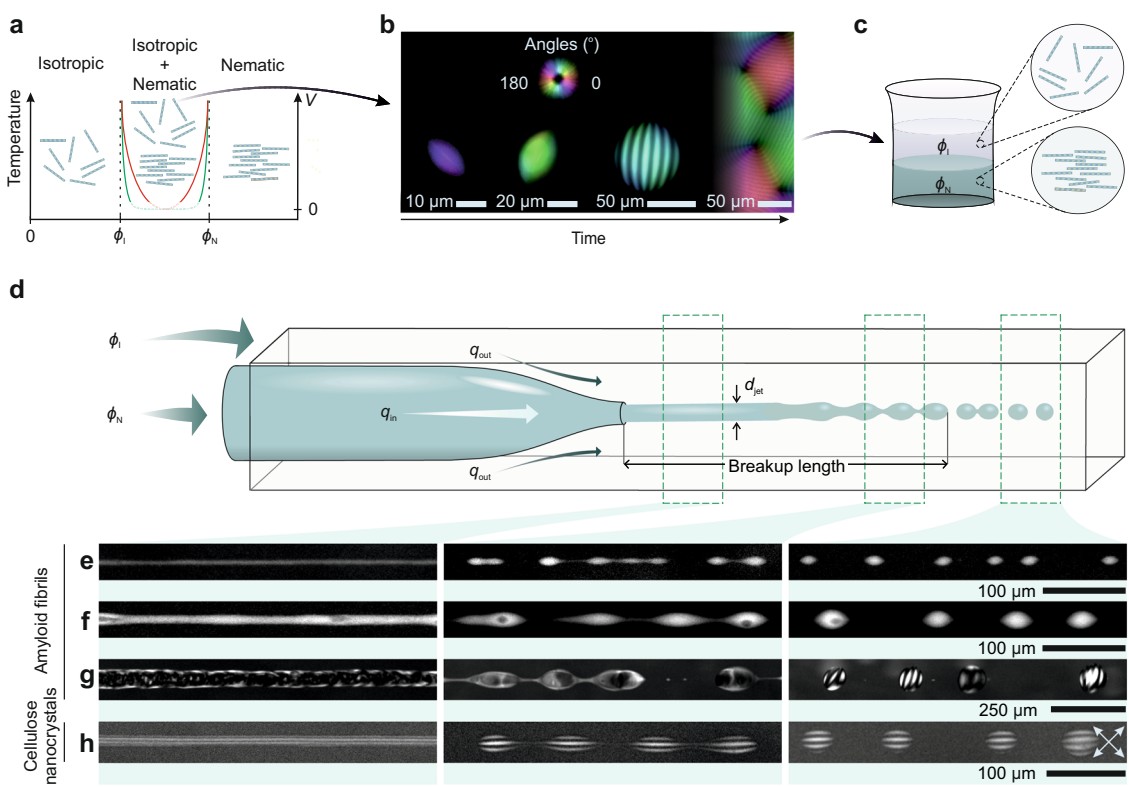

**Fig. 1 | Disentangling kinetics effects from thermodynamics in anisotropic bio-colloidal systems. a** Phase diagram overlaid with volume-composition diagram illustrating the two Onsager branches, $\phi_I$ and $\phi_N$, for anisotropic colloidal systems, along with volume-composition kinetic trajectories of tactoids growing by N&G. The red and green curves sketch, respectively, the volume-composition curves of the growth process of β-lactoglobulin amyloid fibrils and sulfated cellulose nanocrystal systems with an initial concentration within $\phi_I$ and $\phi_N$. The vertical axes denote the temperature for the phase diagram and the volume $V$ for the volume-composition space within which the nematic phase (tactoids) emerge and develop. The volume-composition curves are sketches only: precise trajectories have been theoretically derived elsewhere[26]. **b** LC (liquid crystal)-PolScope images showing the formation of homogenous (first column), bipolar (second column), cholesteric tactoids (third column) and bulk phase (fourth column), through nucleation and growth (N&G), in a solution where the concentration is set within the coexistence region, isotropic + nematic. The colormaps (first row) show the orientation of the director field in the plane. **c** Schematic of a completely phase separated solution initially set at a concentration within the coexistence region, where the nematic phase at concentration $\phi_N$ and the isotropic phase at concentration $\phi_I$ appear at the bottom and on top, respectively. **d** Schematic of the co-flow microcapillary device used to extrude a solution set at one thermodynamic Onsager branch $\phi_N$ inside the other Onsager branch $\phi_I$. The tactoids are formed following jet breakup at breakup length. **e**–**h** Cross-polarized images of the process of the formation of homogenous (**e**), bipolar (**f**), and cholesteric (**g**) tactoids with amyloid fibrils, as well as cholesteric tactoids of cellulose nanocrystals (**h**). To form tactoids with various shapes, volumes, and structures, nematic jets with various diameters $d_{jet}$ (first column) are formed by changing the flow rate of the inner $q_{in}$ and outer fluid $q_{out}$, which is followed by the breakup of the jet (second column) and formation of the tactoids (third column). The crossed arrows denote the crossed polarizers.

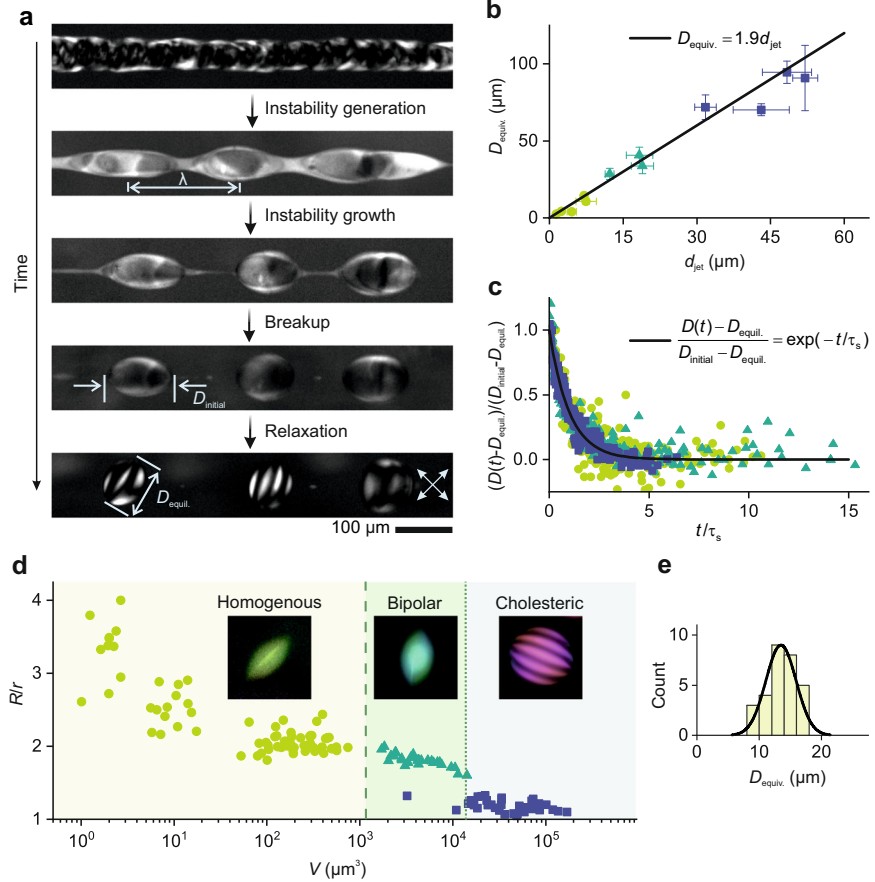

**Fig. 2 | The interplay between fluid flow and thermodynamics rules the dynamics of tactoids formation.** The filled circle, triangle, and square symbols in the plots represent data points from homogenous, bipolar, and cholesteric tactoids, respectively. **a** Rich dynamics of formation of the tactoids captured under the crossed-polarizer: formation of nematic jet, development of Rayleigh-Plateau instability with wavelength $\lambda$, the breakup of jet, and shape and structural relaxation of tactoids with initial length of $D_{initial}$ and equilibrium length of $D_{equil.}$. The crossed arrows denote the crossed polarizers. **b** Experimental data of equivalent diameter $D_{equiv.}$ (=$2V^{1/3}$ where $V = r^2R$ is the scaled volume of tactoids with $R$ and $r$ major and minor axes of tactoids, respectively) as a function of the jet diameter $d_{jet}$. The prediction (solid line) is in good agreement, without fitting parameters, with

experiments showing a linear relationship between $d_{jet}$ and $D_{equiv.}$ for homogenous, bipolar, and cholesteric tactoids. The error bars denote standard deviation. **c** Relaxation behavior is shown by evolution in $(D(t)\text{-}D_{equil.})/(D_{initial}\text{-}D_{equil.})$, with $D(t)$ the length of the tactoids at a given time, as a function of time $t$ parametrized by the shape characteristic relaxation time $\tau_s$. The results from the tactoids with various internal structures and volumes follow a universal curve: $(D(t)\text{-}D_{equil.})/(D_{initial}\text{-}D_{equil.}) = \exp(-t/\tau_s)$. **d** The 3D nematic-cholesteric phase diagram showing the size $V$, shape (aspect ratio $R/r$), and structure of amyloid fibrils tactoids generated by extruding the Onsager branches. **e** Monodispersity of the formed homogeneous tactoids with $D_{equiv.}$ = 14.4 μm showing an 11% coefficient of the variance.

decouple the kinetics effects from thermodynamics with tangible consequences in composition, morphology, shape, structure and size distribution of the ensued tactoids. We bypass the kinetic pathway of the conventional tactoids formation process (N&G/spinodal decomposition), thus replacing the classical N&G time of up to few days for the formation of tactoids with an *induction time* of the order of a few minutes only. By extruding the Onsager branches, we produce tactoids with different aspect ratios compared to those obtained classically and we produce cholesteric droplets maintaining a constant pitch over different volumes, a structural hallmark departing significantly from tactoids obtained by N&G. We finally demonstrate the generality of the approach by creating highly controllable negative tactoids (isotropic within nematic), and multicomponent liquid crystalline droplets based on biocolloids and plasmonic gold nanorods.

Amyloid fibrils and sulfated cellulose nanocrystals are carefully selected as two examples of filamentous colloids undergoing LLCPS via volume-composition curves of different shapes: we expect mild and progressive increase in the trajectory slope for amyloid fibrils (e.g., such as the red line in Fig. 1a), because for this system the periodicity of cholesteric tactoids is known to depend significantly on time and volume[25], a strong enrichment of longer fibrils in the nematic phase is

known to occur during N&G[32] and relaxation dynamic of the internal structure follows a slow kinetics[35]; on the contrary, we expect steep slopes of the volume-composition curve rapidly approaching the Onsager binodal limits for cellulose nanocrystals (e.g., such as the green line in Fig. 1a), since for this system fractionation of long and short fibrils in the nematic and isotropic phase, respectively, is small compared to amyloid fibrils during N&G (see Ref. [32] for the length distributions of isotropic and nematic phases of cellulose nanocrystals and amyloid fibrils, prepared following the same N&G protocol as here), the cholesteric pitch is constant with tactoid volume and the relaxation kinetics is fast[35].

## Results

### Controlling phase separation in microfluidic devices

In our experiments, we use solutions of macroscopically phase separated and equilibrated biocolloids, corresponding to the two Onsager compositions[29], see Fig. 1c and Methods. We then extrude the nematic phase, set at one thermodynamic Onsager composition ($\phi_N$), inside the isotropic phase, set at the other Onsager composition ($\phi_I$), in a two-phase coflowing stream (Fig. 1d) obtaining heterogeneous colloidal systems of controlled droplet sizes, which we then compare with

droplets of identical sizes, but generated by phase separation via N&G (see Methods for concertation values of amyloid fibrils and cellulose nanocrystals). A well-established co-flow microfluidic chip consisting of two co-axially aligned capillary tubes is used to enable three-dimensional co-axial flow of the two phases[36,37]. The solution in $\phi_N$ is injected into the inner cylindrical capillary tube with 0.6 mm and 1 mm inner and outer diameters, respectively. One end of the capillary tube is shaped into tapered orifice with 0.025 mm inner and 0.042 mm outer diameters, respectively. The surrounding medium in $\phi_I$ is injected in the interstices between the inner capillary tube and the outer capillary tube, which is square with 1.05 mm dimension. It is observed that upon injection of the solution using syringe pumps, the nematic jet is formed first; downstream the instability appears on the surface of the jet, which ultimately leads to the breakup of the jet into droplets at breakup length (Fig. 1d). The jet diameter $d_{jet}$ is controlled easily by changing the flow rate of the inner $q_{in}$ and outer fluid $q_{out}$. However, care should be taken, since due to the vanishingly low surface tension of the tactoids[25,38] -$10^{-7}$–$10^{-6}$ N m$^{-1}$, there is an upper limit for the flow rate, above which the tactoids are not able to keep their equilibrium shape and structure. Another upper limit for the flow speed is when the orientational order is induced by the flow field resulting in isotropic-to-nematic transition in the medium phase as pointed by de Gennes[39]. We performed our experiments below the mentioned limits where $q_{in}$ was maximum 5 μL h$^{-1}$ and $q_{out}$ maximum 50 μL h$^{-1}$.

Amyloid fibrils tactoids with various shapes, sizes, and structures are formed by simply controlling the flow parameters (see Fig. 1e–g, Supplementary Fig. 1, and Supplementary Movies 1–7). At a low $q_{in}/q_{out}$ ratio of $7 \times 10^{-5}$, a nematic jet with $d_{jet} = 7.0$ μm is generated, which upon breakup results in the formation of homogenous tactoids with equivalent diameter $D_{equiv.}$(=$2V^{1/3}$ where $V = r^2R$ is the scaled volume of tactoids with $R$ and $r$ major and minor axes of tactoids, respectively) of $14.4 \pm 1.6$ μm (Fig. 1e and Supplementary Movie 3). At $d_{jet}$ of 12.3 μm, achieved at $q_{in}/q_{out} = 23 \times 10^{-5}$, we observe the formation of tactoids with a larger volume with $D_{equiv.}$ of $28.5 \pm 3.6$ μm and bipolar structure (Fig. 1f and Supplementary Movie 4). At higher $q_{in}/q_{out} = 335 \times 10^{-5}$, a nematic jet with $d_{jet} = 48.3$ μm is generated which upon breakup leads to the formation of droplets ($D_{equiv.} = 94.5 \pm 7.3$ μm) with cholesteric internal structure (Fig. 1g and Supplementary Movie 7).

When performing experiments with cellulose nanocrystals (see Methods), cholesteric tactoids with $D_{equiv.} = 73.6 \pm 10.3$ μm are formed similarly to cholesteric amyloid fibril tactoids (Fig. 1h). Strikingly, we observe the formation of cholesteric structure in cellulose nanocrystal system already within the nematic jet where the cholesteric pitch of the jet is 12.3 μm, slightly lower than the one observed for the cellulose nanocrystals tactoids formed with our approach, which have a pitch value of 14.6 μm. This is different from the amyloid fibrils system, implying that the nematic jet of the cellulose nanocrystals relaxes to the cholesteric structure before the breakup, which can be related to the fast relaxation dynamics of cellulose nanocrystals system compared to amyloid fibrils[35].

## The interplay between fluid flow and thermodynamics

Our experiments reveal how the interplay between fluid flow and thermodynamics rules the dynamics of tactoid formation, leading to a rich behavior in the formation of the tactoids from the nematic liquid crystalline jet (Fig. 2a). Upon the formation of the nematic jet, driven by Rayleigh-Plateau instability, the jet is axi-symmetrically perturbed downstream. The perturbation on the nematic jet grows by time and ultimately leads to a breakup of the jet into a chain of elongated droplets. This is followed by shape and structural relaxation of the droplets to their equilibrium state driven by free energy minimization where both liquid crystalline anisotropic and isotropic free energy terms contribute (see below). Another remarkable feature of our approach is the time scale of the tactoids formation, i.e., the *induction time*, which, as opposed to the N&G pathway taking minutes to days, is

in the order of a few minutes only (Supplementary Figs. 2–3 and Supplementary Movies 1–7). This noteworthy reduction in the induction time is rooted in the bypass of the kinetic pathway of the conventional tactoids formation process (N&G/spinodal decomposition), allowing forming tactoids of final composition, structure, and dimension independently from the limiting parameters of the N&G[40], such as the thermodynamic energy barrier of nucleation and the finite transport rates of the growth phase.

To establish a quantitative description of the size of the tactoids formed from a nematic jet with a given $d_{jet}$, we approximate the volume of the tactoids to be the volume contained in one wavelength $\lambda$ of the instability appearing in the nematic jet, $V = \pi d_{jet}^2 \lambda/4$, where, for simplicity, we refer to the undisturbed stage of the jet still in its transient cylindrical shape of diameter $d_{jet}$. We consider the most unstable mode of the Rayleigh-Plateau instability[36,37,41,42], which depends on the viscosity ratio of the nematic-to-isotropic phases; with a viscosity ratio of 0.5 (Ref. [38]), we obtain $\lambda = 4.6d_{jet}$ (Ref. [42]). Thus, we compute $D_{equiv.} = 1.9d_{jet}$, predicting well and without fitting parameters the linear relation found in the experiments, as shown in Fig. 2b. Note that, although the viscosity of the nematic phase changes depending on the director field orientation, for simplicity we use the zero shear viscosity values for both isotropic and nematic phases in our analysis. In Supplementary Note 1, we show that the dissipative energy related to Rayleigh-Plateau instability is several orders of magnitude lower than the thermodynamic energy of the system, and thus can be neglected in what follows.

The shape relaxation behavior of the elongated droplets to the equilibrium state, following the breakup, shows a first-order exponential decay following a universal curve of $(D(t)-D_{equil.})/(D_{initial}-D_{equil.}) = \exp(-t/\tau_s)$, with $D(t)$, $D_{initial}$ and $D_{equil.}$ the length of the tactoid at a given time $t$, at the initial time and at equilibrium state, respectively (Fig. 2c). The time $t$ has the zero point at the point of pinch-off of the jet into the tactoids (Fig. 2a) and is parametrized by the shape characteristic relaxation time $\tau_s$ that is determined by fitting $(D(t)-D_{equil.})/(D_{initial}-D_{equil.}) = \exp(-t/\tau_s)$ to every single tactoid shape relaxation data set (see Supplementary Fig. 4). These results show similar relaxation behavior as in our recent work on artificially elongated tactoids (see Ref. [43] for detailed analysis). Furthermore, once formed, all tactoids are oriented in the flow direction, except for cholesteric amyloid fibrils tactoids with four or higher number of bands, which are orientated almost perpendicular to flow direction, a behavior whose underpinning physics is not entirely clear yet, calling for further investigation (see Fig. 1e–h and Supplementary Fig. 1).

The 3D tactoidal phase diagram capturing the size $V$, shape (aspect ratio $R/r$), and structure of the amyloid fibrils tactoids formed using our approach is presented in Fig. 2d. We can generate tactoids with various volumes, ranging from 1.0 μm³ to 169,491.4 μm³, aspect ratios, and internal structures. This expands significantly the scope of microfluidics in liquid crystalline systems that was limited, previously, to confining a nematic phase inside another immiscible fluid (e.g., oil), resulting only in spherical droplets and with only parallel anchoring due to the dominating interfacial tension[44,45]. We find the tactoids generated with our approach fairly monodisperse despite the low interfacial tension involved, with a coefficient of the variance of 11% for tactoids with $D_{equiv.} = 14.4$ μm, formed by a jet with $d_{jet} = 7.0$ μm (Fig. 2e).

## Negative tactoids formation by extruding Onsager branches

Next, we take advantage of the possibilities offered by this approach to form the classes of tactoids known as negative tactoids or atactoids[46–48]. These tactoids are understood to be microdroplets of the isotropic phase within the nematic phase and have been poorly studied[46]. To this end, we use the same microfluidic system as shown in Fig. 1d, but we extrude the isotropic phase set at the Onsager composition $\phi_I$, inside the nematic phase set at $\phi_N$ (see Fig. 1a). We note

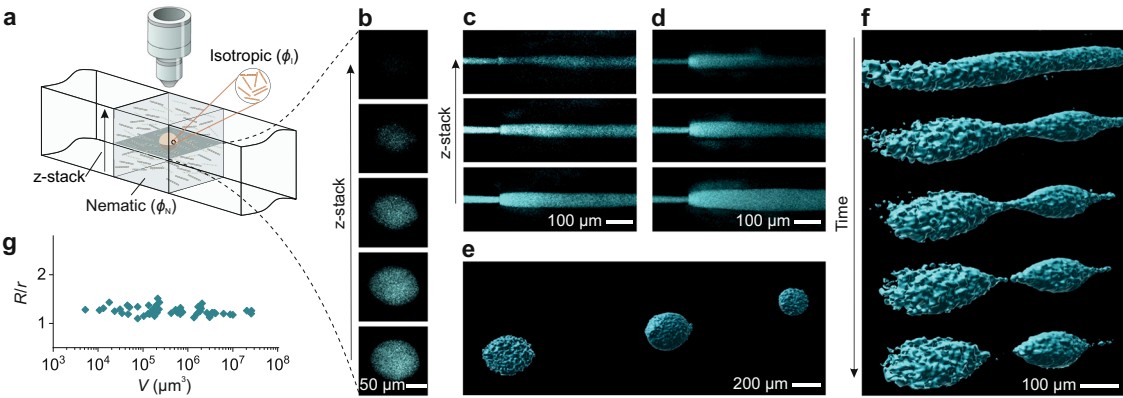

**Fig. 3 | Negative tactoids formation by extruding the Onsager branches.**
**a** Schematic of an isotropic droplet, that is set at one thermodynamic Onsager branch $\phi_I$, within a nematic phase set at the other Onsager branch $\phi_N$, in the microfluidic system, where the fibrils in the isotropic phase have been fluorescently dyed with Thioflavin T. **b** Confocal images of a single isotropic droplet in different z-stacks, showing the visibility of the tactoid at different heights. **c,d** Confocal images of isotropic jet with different diameters close to the tip of the inner capillary tube at different z-stacks. **e** 3D reconstructed image of three negative tactoids generated by extruding the Onsager branches. **f** 3D reconstructed images showing the dynamics of the negative tactoids formation: formation of the negative jet, the appearance of the instability, the breakup of the jet, relaxation of the negative tactoids. **g** Phase diagram of the negative tactoids formed by extruding the Onsager branches showing aspect ratio of tactoids versus their volumes.

that the birefringence of the surrounding nematic medium makes impossible to distinguish the embedded isotropic tactoids under crossed polarizers, and therefore, the isotropic phase was dyed[49] with Thioflavin T before extrusion and imaged by confocal fluorescence microscopy (see Fig. 3a, b, and Methods). A jet of the isotropic phase is generated with different diameters by changing the flow rate (Fig. 3c, d). A clear boundary between isotropic and nematic phase is observed. A series of isotropic tactoids inside the nematic medium is formed (Fig. 3e), where as 3D reconstructed images in Fig. 3f and Supplementary Movie 8 illustrate, formation dynamics show similar steps as in the case of nematic tactoids within the isotropic phase. The phase diagram of the negative tactoids, showing the size and shape range, is given in Fig. 3g; note that the tactoids with small sizes are not identified which may be due to a very weak emitted fluorescence signal, making it undetectable. Yet, our results show that, unlike direct tactoids, negative tactoids of different sizes hold nearly constant aspect ratio over four decades of volumes, which we show in the supporting information to be a consequence of the flow-induced alignment in the medium phase (Supplementary Note 2 and Supplementary Fig. 5). In addition, the aspect ratio of $1.25 \pm 0.08$ is preserved up to volumes well beyond $10^7 \, \mu m^3$, sizes at which direct tactoids typically exhibit an aspect ratio of 1 (see Fig. 4a for comparison).

### Decoupling kinetics from thermodynamics effects

Disentangling kinetics from thermodynamics effects allows revealing the emergence of physical phenomena not observed before. To highlight this, we carry out a comprehensive physical analysis on the tactoids formed with our approach and compare them to control experiments achieved via phase separation realized by classical N&G. The control experiments are done by triggering phase separation via N&G in two different ways: either diluting a system initially set at $\phi_N$ into the coexistence region, or by allowing direct N&G of tactoids from a composition falling within the 2-phase region, but previously destabilized by vortexing (Fig. 4a). While all three pathways show similar behavior of decreasing aspect ratio with an increase in volume, when the tactoids are formed by extruding the Onsager branches several new fundamental differences appear.

First, amyloid fibril homogeneous tactoids of smaller volumes become detectable by our method (Fig. 4a). This is a direct consequence of the energy-activated nucleation process, which sets a minimum observable size for nuclei of the tactoids to be stable[26,40], whereas extrusion of the Onsager branches extends, virtually, to

vanishingly small sizes (see Fig. 4a). Secondly, homogeneous tactoids with smaller aspect ratio appear, which is also inaccessible to the other two N&G paths. The reason is that the aspect ratio is directly linked to the average length of constituent particles[33,50] which is higher for the tactoids formed by N&G compared to those formed with our approach, since N&G selectively fractionates longer from shorter fibrils during phase separation, which is particularly true in the amyloid fibrils case[32,51]. Note that no change is observed over time in the structure of the tactoids generated by microfluidic breakup, indicating that the tactoids are essentially forming already in the final equilibrium configuration (Fig. 4a).

Thirdly, it becomes now even possible to highlight differences in the internal structure of tactoids achieved via N&G and by extruding the Onsager branches. To this end, we explore the relation of the cholesteric pitch with the volume of the tactoids in both the amyloid fibrils and cellulose nanocrystals case. Previously, it was observed that the pitch decreases for amyloid fibrils cholesteric tactoids as the volume grows during N&G[25,26]. This arises from the change in the concentration of the tactoids during the growth phase, and it is generally well known that the pitch decreases with an increase in the concentration of the building blocks in the solution[52–54]. Remarkably, for amyloid fibril cholesteric tactoids formed by extruding the Onsager branches, we observe a pitch value of $46.1 \, \mu m$ which stays constant over a broad range of tactoid volume sizes (Fig. 4c, d, and Supplementary Fig. 6). In contrast, the same amyloid fibrils tactoids produced by N&G feature, as in previous reports[25], a steady decrease of the pitch with tactoid volume over the same range of sizes, until reaching a plateau value of $16.7 \, \mu m$. The remarkable difference in pitch size observed in the two cases and the size-independence of the pitch when N&G is avoided, point at $46.1 \, \mu m$ as the true equilibrium pitch (see Supplementary Fig. 7), which would then be only observable by extruding the Onsager branches and highlight the challenges associated to the measurements of natural pitch on macroscopically separated isotropic and nematic phases. To consolidate this point further, we first measure the natural pitch independently in negative tactoids obtained by reverse Onsager extrusion and we find again $46.1 \, \mu m$ (see Supplementary Fig. 7). Secondly, we note that on similar systems the measurement of the natural pitch in the absence of the gravity on horizontally laid capillaries confirms a natural pitch in the order of $43 \, \mu m$ (Ref. [35]). Thus, we infer that the pitch of $16.7 \, \mu m$ measured by N&G and on vertically laid cuvettes is not the fully relaxed thermodynamic equilibrium pitch due to non-equilibrium and

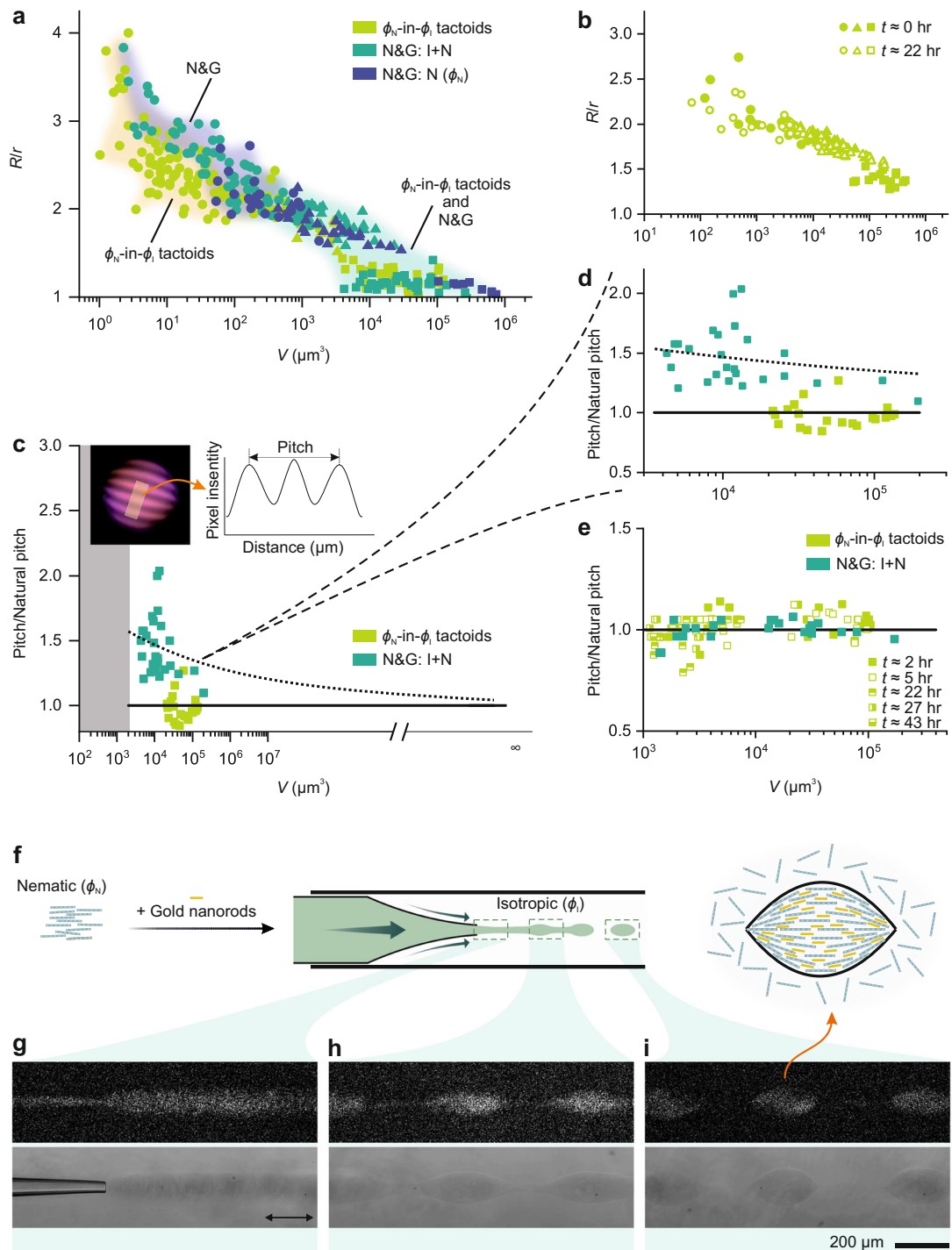

**Fig. 4 | Disentangling kinetics from thermodynamics effects reveals unprecedented physical phenomena.** The circle, triangle, and square symbols in plots represent data points from homogenous, bipolar, and cholesteric tactoids, respectively. Filled symbols denote the tactoids captured right after the relaxation, while unfilled symbols represent long-time measurements. **a** Phase diagram of tactoids obtained by extruding the Onsager branches ($\phi_N$-in-$\phi_I$ tactoids) versus two control experiments following N&G. **b** Aspect ratio versus volume of amyloid fibrils tactoids formed by extruding the Onsager branches and tracked over time. **c** Pitch of cholesteric tactoids of amyloid fibrils obtained via N&G and by extruding the Onsager branches ($\phi_N$-in-$\phi_I$ tactoids). While the pitch value decreases with an increase in volume for tactoids formed via N&G, it remains constant for tactoids formed by extruding the Onsager branches. Pitch is renormalized by the observable bulk pitch of the system which is 16.7 μm for N&G from a composition falling within isotropic + nematic region and 46.1 μm for tactoids formed by extruding the Onsager branches, calculated as an average value of the pitch of tactoids. (Pitch/

natural pitch −1) versus tactoid size remains constant (full line) for the thermodynamic case, but decreases as $-V^n$ for N&G (dashed line), with $n$ a fitting parameter equal to −0.17. **d** The inset shows the detailed evolution of the pitch versus tactoids volume. **e** Pitch of cholesteric droplets from cellulose nanocrystals compared to those obtained via N&G pathways and by extruding the Onsager branches ($\phi_N$-in-$\phi_I$ tactoids). Tracking pitch of tactoids over long time shows no significant changes. Natural pitch is measured to be 15.9 μm for N&G and 14.6 μm for tactoids obtained by extruding the Onsager branches. **f–i** Extruding the nematic phase mixed with gold nanorods into the isotropic phase (**f**) results in the formation of the hybrid jet (**f**) driven by instability (**h**) breaking to chain of hybrid amyloid fibrils-plasmonic gold nanorods tactoids (**i**). The samples are excited with linearly polarized light and collecting the confocal fluorescence signal (top row) or the transmitted light (bottom row). The polarization direction of the excitation laser is denoted with a black double arrow.

gradient effects (concentration, size, sedimentation, gravity). This shows the possibility to control the internal structure of tactoids beyond what is possible with N&G and allows us to experimentally confirm, for the first time, the theoretical rationale which has been advanced to explain the dependence between internal periodicity and volume of cholesteric tactoids during their growth[26]. Note that Fig. 4c is extended to $V \to \infty$ as the pitch value is measured experimentally for the bulk phase having followed the N&G pathway.

Since the deviation between equilibrium and morphologies observed by N&G is understood to arise from the gentle progressive increase of the slope of the volume-composition curve toward the Onsager boundaries, we expect a completely different behavior when the trajectory is steeply and rapidly approaching the two asymptotic boundaries; in line with our expectation, and in stark contrast with amyloid fibrils case, the cellulose nanocrystal cholesteric pitch remains constant, with no change in the internal structure of the tactoids over the same range of volume considered, and this independently of whether the tactoids are formed via N&G or by extruding the Onsager branches (Fig. 4e and Supplementary Fig. 6): the proximity between the volume-composition curves and the Onsager asymptotes smear in the case of cellulose nanocrystals the composition and structural differences in the tactoids obtained by the two methods. Accordingly, the pitch of the cellulose nanocrystals tactoids formed by N&G or extruding the Onsager branches is matching within the experimental error (Supplementary Fig. 6) and found totally independent of time (Fig. 4e).

The possibility of controlling the structure of heterogeneous liquid crystals beyond state of the art, can also be extended to the generation of complex multicomponent colloidal fluids. As an example, we fabricate hybrid amyloid fibril-plasmonic gold nanorod tactoids by extruding a nematic phase (set at the binodal $\phi_N$) that is mixed with guest gold nanorods, in an isotropic medium (at binodal line $\phi_I$) (Fig. 4f–i). The nanorods follow the alignment of the host fibrils liquid crystal in the nematic jet and the ensued tactoids, as evident in the fluorescence images with polarization sensitivity (Fig. 4i). The strong fluorescence signals in the images in Fig. 4g–i, also show that the concentration of the nanorods remains high when the tactoids are formed[55]. This technique produces the hybrid tactoids within minutes, remarkably faster than the days needed by the existing methods[55,56], and allows full control over the concentration of the guest particles within the host amyloid tactoids, which is critically important for plasmonic applications, for enhancing fluorescence of amyloid fibrils, or for controlling the alignment and spatial distributions of gold nanorods within host amyloid fibrils or cellulose nanocrystals[55,56].

## Discussion

In closing, we have shown that it is possible to disentangle kinetics from thermodynamics effects in heterogeneous colloidal systems. We have shown that our approach offers original possibilities in the formation of liquid crystalline droplets with significant reduction in induction time, expanded phase diagram, and control over their composition, morphology, shape, structure, and size distribution. We have further demonstrated that it is possible to form on-demand negative tactoids, that is, isotropic droplets in a nematic medium, and multicomponent liquid crystalline droplets with precision control of their size and composition. These findings deepen our fundamental understanding of the interplay between kinetics and thermodynamics in the formation of heterogeneous colloidal systems and may offer unexplored possibilities in all those technological applications relying on N&G as a sole mechanism to establish the final structure and morphology.

## Methods
### Preparation of suspensions of amyloid fibrils
The amyloid fibrils suspension was prepared from β-lactoglobulin purified from whey protein, see detailed protocol in Ref. [32]. A suspension of 2 wt.% β-lactoglobulin in 300 mL Milli-Q water was prepared. The suspension was filtered through a 0.45 μm nylon syringe filter (Huberlab) and the pH of the suspension was adjusted to 2 by adding HCl. To form amyloid fibrils, the solution was then heated at 90 °C for 5 h. This was followed by applying mechanical shear forces to shorten the length of the fibrils. To remove unreacted monomers and peptides from the suspension, dialysis was run for 5 days using a 100 kDa (MWCO) Spectra/Por dialysis membrane (Biotech CE Tubing) with daily bath replacement. The desired concentration for the suspension was achieved by up-concentrating with reverse osmosis method using a Spectra/Por 1 dialysis membrane (standard RC tubing) against a 10 wt.% polyethylene glycol solution (Sigma Aldrich) pH 2. After the complete macroscopic phase separation, the concentrations of the two Onsager branches were measured to be $\phi_I = 2.0 \pm 0.1$ wt% and $\phi_N = 2.5 \pm 0.2$ wt%.

### Preparation of suspension of cellulose nanocrystals
As described in Ref. [32], 2.5 wt.% of freeze-dried cellulose nanocrystal (FPInnovations) was dissolved in milliQ water and sonicated for 120 s. To remove unwanted particles, the solution was centrifuged at $12,000 \times g$ for 20 min. For cellulose nanocrystals, we measured the concentrations of the two Onsager branches to be $\phi_I = 2.4 \pm 0.1$ wt% and $\phi_N = 2.9 \pm 0.2$ wt%.

### Microfluidic device
The microfluidic device consisted of the two co-axially aligned capillary tubes: A square borosilicate glass capillary tube (VitroCom) with 1.05 mm inner dimension and a cylindrical borosilicate glass capillary tube (Hilgenberg) with inner and outer diameters of 0.60 mm and 1.00 mm, respectively. One end of the cylindrical capillary was shaped into a tapering orifice with inner and outer diameters of 0.025 mm and 0.042 mm at orifice tip, respectively. The round capillary was inserted into the square capillary. The capillaries were assembled on a glass slide (Thermo Scientific) with the help of Bondic liquid plastic welder.

One end of the inner and the outer capillary tubes was connected to a blunt needle with 0.34 mm inner diameter and 0.64 mm outer diameter. A flexible tube with 0.8 mm inner diameter is used to connect the syringes (Hamilton), filled with the solutions, to the blunt needles. Syringe pumps (PhD 2000, Harvard Apparatus) are used to inject the solutions into the microcapillary device.

### Experimental details
To ensure that there were no contaminants in the system, all capillaries, tips and syringes were cleaned with ethanol and to avoid tip blockage in the capillary tubes, the solutions were centrifuged before each experiment. In our experiments, great care was taken to prevent backflow, which may lead to a mixture of two phases. For instance, when starting the experiments, the syringe pumps were run in a way to move the front edge of both jet and the medium phase at the same time; later the flow rates were adjusted to the desired rate. All experiments were performed at room temperature.

### Sample characterization
Four different microscopy systems were used to characterize different types of samples involved. First, cross-polarized optical microscopy (Zeiss Axio Imager Z2) with an attached camera (AxioCam MRc) with 5× (Achrostigmat), 10× (Plan Neofluar), 20× (Epiplan-Neofluar) and 50× (Epiplan-Neofluar) objectives was used to record the images of the tactoids. The images were acquired under time-series mode at frame rates between 300 and 600 frames per minute depending on the speed of the flow inside the microfluidic system. Note that the microfluidic device was placed with 45° with respect to one of the polarizers of the microscope, allowing us to detect the jet and tactoids clearly. Second, a microscope (Zeiss Axio Imager M1m) equipped with LC (liquid crystal)-PolScope and objectives of 5× (Achrostigmat), 10×

(Plan Neofluar), and 20× (Epiplan-Neofluar) was used to capture the PolScope images of the tactoids. Third, confocal microscopy (Zeiss LSM 780 Axio Imager 2) with 10× objective (EC Plan-Neofluar) was used to record the results reported in Fig. 3. All analyses were performed using the ImageJ and/or Zen software. Forth, in the experiments of hybrid amyloid fibrils-plasmonic gold nanorods tactoids, a confocal microscope (Leica TCS SP8) with a 10× objective (HC PL Fluotar Ph1) is used to acquire the polarization-dependent fluorescence images. The excitation source used in our experiments was a continuous laser at 633 nm with the power of 4 mW.

### Preparation of amyloid fibrils dyed with Thioflavin T
In the case of the negative tactoids experiments, the isotropic phase was stained with the fluorophore Thioflavin T with mix ratio of 100:1 by volume. Thioflavin T binds specifically to the β-sheets of the amyloid fibers and shows enhanced fluorescence. In our microscopy measurements, the excitation was set at 458 nm wavelength and the detection wavelength between 463 and 553 nm.

### Gold nanorods experiments
Gold nanorods are prepared following the protocol described in Ref. [55]. In our experiments, we mixed 40 μL of the highly concentrated gold nanorods dispersion with 200 μL of the nematic phase of amyloid fibrils solution. Then, the mixture was filled in the co-flow microfluidic chips and imaged with fluorescence microscopy as described above.

## Data availability
The data that support the findings of this study are available from the corresponding author upon request.

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

## Acknowledgements

We thank T. Schwarz and S. S. Lee from the Scientific Center for Optical and Electron Microscopy of ETH Zurich (ScopeM) for the help with the polarized optical microscopy and the laser scanning confocal micro-scopy and for the technical support with microfluidic experiments. We also thank Q. Sun, I. Kutzli, J. Zhou, and J. Vogt for technical support and discussions. Support from the Swiss National Science Foundation—Sinergia Scheme Grant No. CRSII5_189917—is gratefully acknowledged (R.M. and P.F.).

## Author contributions

H.A. and R.M. conceived the idea and initiated the project, designed the experiments, and performed theoretical interpretation of the results and the theoretical analysis underpinning the project. H.A. and S.M. built the experimental apparatus, performed experiments, and analyzed the data. H.A. and Y.Y. performed gold nanorods experiments and analyzed the data. P.F. contributed to the experiments. R.M. supervised the research. H.A., S.M., and R.M. wrote the paper with input from all authors.

## Competing interests

The authors declare no competing interests.
