## [Peer Review File · Nature Communications]

Disentangling kinetics from thermodynamics in heterogeneous colloidal systemsEditorial Note: This manuscript has been previously reviewed at another journal that is not operating a transparent peer review scheme. This document only contains reviewer comments and rebuttal letters for versions considered at Nature Communications.

Reviewers' Comments:

Reviewer #1:

Remarks to the Author:

The authors have adequately addressed the questions that arose during the last round of review for the Nature Sister journal. I believe the manuscript is considerably improved and should represent a meaningful advance to the field.

I recommend publication in Nature Communications.

Reviewer #2:

Remarks to the Author:

The revised version of the manuscript entitled "Disentangling kinetics from thermodynamics in heterogeneous colloidal systems" has been significantly improved and I found that the authors have answered to my concerns.

Therefore I suggest publication of the manuscript without any further revision from my side.

Reviewer #1

The authors have adequately addressed the questions that arose during the last round of review for the Nature Sister journal. I believe the manuscript is considerably improved and should represent a meaningful advance to the field.

I recommend publication in Nature Communications.

We appreciate the Reviewer's thoughtful and constructive comments, which helped us to improve our manuscript.

Reviewer #2

The revised version of the manuscript entitled "Disentangling kinetics from thermodynamics in heterogeneous colloidal systems" has been significantly improved and I found that the authors have answered to my concerns.

Therefore I suggest publication of the manuscript without any further revision from my side.

We appreciate the Reviewer's thoughtful and constructive comments, which helped us to improve our manuscript.